# Analysis of Furan and Its Derivatives in Food Matrices Using Solid Phase Extraction Coupled with Gas Chromatography-Tandem Mass Spectrometry

**DOI:** 10.3390/molecules28041639

**Published:** 2023-02-08

**Authors:** Wen-Xuan Tsao, Bing-Huei Chen, Pinpin Lin, Shu-Han You, Tsai-Hua Kao

**Affiliations:** 1Department of Food Science, Fu Jen Catholic University, New Taipei City 242, Taiwan; 2National Institute of Environmental Health Sciences, Miaoli 350, Taiwan; 3Institute of Food Safety and Risk Management, National Taiwan Ocean University, Keelung 202, Taiwan

**Keywords:** furan and its derivatives, SPME arrow, GC-MS/MS

## Abstract

The objective of this study was to develop a simultaneous analysis method of furan and its 10 derivatives in different food commodities. The results indicated that furan and its 10 derivatives could be separated within 9.5 min by using a HP-5MS column and gas chromatography–tandem mass spectrometry (GC-MS/MS) with multiple reaction monitoring mode for detection. Furthermore, this method could resolve several furan isomers, such as 2-methyl furan and 3-methyl furan, as well as 2,3-dimethyl furan and 2,5-dimethyl furan. The most optimal extraction conditions were: 5 g of the fruit or juice sample mixed with 5 mL of the saturated NaCl solution, separately, or 1 g of the canned oily fish sample mixed with 9 mL of the saturated NaCl solution, followed by the equilibration of each sample at 35 °C for 15 min, using a carboxen-polydimethylsiloxane SPME arrow to adsorb the analytes for 15 min at 35 °C for subsequent analysis by GC-MS/MS. For method validation of all the analytes in the different food matrices, the recovery was 76–117% and the limit of the quantitation was 0.003–0.675 ng/g, while the relative standard deviation (RSD%) of the intra-day variability range from 1–16%, and that of the inter-day variability was from 4–20%. The method validation data further demonstrated that a reliable method was established for the analysis of furan and its 10 derivatives in commercial foods.

## 1. Introduction

Furan (C_4_H_4_O) is a liposoluble, highly volatile liquid with a low boiling point. It and one of its derivatives, furfuryl alcohol, are listed by the International Agency for Research on Cancer as Group 2B (possibly carcinogenic to humans), whereas another derivative, furfural, is listed in Group 3 (not classifiable as to its carcinogenicity to humans) [1]. Other derivatives, including methylfuran, ethylfuran, and furfuryl alcohol, are suspected to be genotoxic and can potentially damage the lung, kidney, and liver [2,3,4].

Various substances can be a precursor of furan when heated under high temperature, including carbohydrate, protein, vitamin C, and fat. Accordingly, furan and its derivatives are often present in food products such as coffee, canned food, and baby food, and they can be generated through the following pathways: (1) Maillard reaction or nonenzymatic browning, as well as the thermal degradation and rearrangement of carbohydrates, which produce aldotetrose derivatives, leading to furan formation when cyclized [5]; (2) formation of acetaldehyde and glycolaldehyde from amino acids during the thermal degradation or decarboxylation, dehydration, and Strecker degradation, which in turn produced furan following further condensation and cyclization [6]; (3) oxidative or nonoxidative degradation of ascorbic acid derivatives, such as xylosone, 2-deoxyaldotetrose, and 3-deoxypentosulose, resulting in furan production after cyclization [7]; (4) cyclization and dehydration of the intermediate products of the oxidation of polyunsaturated fatty acids or carotenoids, leading to furan production [7]; and (5) reaction of the ionizing-radiation-induced free radicals with components of food products to produce furan [8]. As stated above, food is easy to form furan during various heating processes, therefore, it is very important to understand the content of furan in food.

For the analysis of the single compound furan, the HP-PLOT Q column (15 m × 0.32 mm, 20 μm) [9] and the BPX-volatiles (60 m × 0.25 mm, 1.4 μm) [10] were the most commonly used GC columns in the literature. By contrast, the columns for furan derivatives include the HP-PLOT Q column (30 m × 0.32 mm, 20 μm), which was used to analyze furan and six of its derivatives (2-methylfuran, 2-ethylfuran, 2,5-dimethylfuran, 2-propylfuran, 2-butylfuran, and 2-pentylfuran) within 40 min, but failing to separate 2,5-dimethylfuran and 2-ethylfuran [11]. The CP-Wax 52 CB column (60 m × 0.25 mm, 0.25 μm), for which 30 min was required to analyze the furan, 2-methylfuran, 2-pentylfuran, 2-butylfuran, 2-acetylfuran, furfural, and furfuryl alcohol in baby food, but that failed to separate the 2,5-dimethylfuran and 2-ethylfuran [12]. The HQ-PLOT Q column (15 m × 0.32 mm, 0.25 μm) successfully separated 2-methylfuran and 3-methylfuran, as well 2-ethylfuran and 2,5-dimethylfuran, but only detected five analytes and spent 30 min on separation [13]. These GC columns include ones with high polarity and ones with low polarity, and the number of analytes they are able to detect is unsatisfactory. Therefore, further evaluation is required.

As furan and its derivatives are volatile and present in food products in trace concentrations, they are often extracted using headspace extraction [8] or solid-phase microextraction (SPME) [13] fiber, followed by the analysis by gas chromatography-mass spectrometry (GC-MS). In many published reports, only furan and its six derivatives have been analyzed in food products, due to the difficulty in separating the furan isomers, such as 2-methylfuran and 3-methylfuran, or 2,5-dimethylfuran and 2,3-dimethylfuran [11,12]. Additionally, the equilibration time, extraction time, and separation time is lengthy, which takes about 1 h to complete analysis [11,12].

Compared with the traditional SPME fiber, the SPME arrow used in our study can provide a larger adsorption area, with an arrow-shaped tip making it easier to pierce the septum of the sample vail, which cannot only extend the life of the sorption material, but also reduce the detection limit and improve recovery and reproducibility. More importantly, the SPME efficiency will be affected by its materials and interactions between absorbent with analytes. For example, polydimethylsiloxane with low polarity and carbowax polyethylene glycol with high polarity can adsorb different components. The π bonds, hydrogen bonds, or van der Waals forces are the common interactions existing in the SPME extraction process [14]. However, no information is available to compare the variety of the SPME arrow and extraction condition in the extraction of furan and its derivatives for different food commodities, while only few studies used tandem mass spectrometry (MS/MS) for the analysis of furan and its derivatives. In view of the complicated matrix of various food products and the similarity of furan’s molecular fragments to those of other volatile gases, it is necessary to develop an SPME arrow-GC-MS/MS method for the simultaneous determination of furan and its 10 derivatives in commercial food products.

## 2. Results and Discussion

### 2.1. Evaluation of Simultaneous Separation Using GC-MS/MS

#### 2.1.1. Evaluation of the Column

In our study, both HP-WAX column (30 m × 0.25 mm, film thickness 0.5 μm, Agilent, Santa Clara, CA, USA) and HP-5MS column (30 m × 0.25 mm, film thickness 0.25 μm, Agilent, Santa Clara, CA, USA), were selected for the evaluation of the separation efficiency. With helium as the carrier gas and flow rate at 1 mL/min, the HP-WAX column was able to separate furan and its 8 derivatives, excluding 2-ethylfuran and 2,5-dimethylfuran with 10 min. However, tailing occurred in the furan peak and the other five furan derivative peaks (Figure 1A).

By using an HP-5MS column instead, with a split ratio at 1:10 and the split flow rate at 10 mL/min, furan and its 10 derivatives were adequately separated within 10 min with the inlet temperature at 280 °C, following the temperature programming conditions of 32 °C in the beginning, maintained for 4 min, raised to 200 °C at 20 °C/min, and maintained for 3 min. Most importantly, the furan isomers, including 2-methyl furan and 3-methyl furan, as well as 2,5-dimethylfuran and 2,3-dimethylfuran, were adequately resolved (Figure 1B).

The HP-5MS column has been used for the separation of furan and its derivatives in other manuscripts, however, only four components, including furan, furfural, 2-methylfuran, and 2-pentylfuranr, were analyzed and had an asymmetrical peak shape [15]. In our study, not only a total of furan and its nine derivatives could be separated in a similar time, but also good resolution maintained for each derivative, including isomers, while the good resolution facilitates more accurate identification than the previous research [11,12]. From the viewpoint of the peak resolution (R) in chromatography, a high resolution of two was reached for 2-methylfuran and 3-methylfuran, as well as for 2,5-dimethylfuran and 2,3-dimethylfuran, in our study. In addition, a resolution of one was shown for 2-ethylfuran and 2,5-dimethylfuran. By contrast, comparatively, the resolution value obtained in this study was higher than that reported by Frank et al. [13], showing a resolution of 2,5-dimethylfuran and 3-methylfuran, as well as 2-ethylfuran and 2,5-dimethylfuran being 1.25 and 0.57, respectively. Our finding further demonstrated a high resolution efficiency of the separation condition developed in our study.

#### 2.1.2. Ion Selection of MS/MS

Table 1 presents the MS/MS parameters and retention time of furan and its 10 derivatives, as well as the internal standard by MRM mode, while Figure 2 presents the MRM chromatograms of furan and its 10 derivatives, using the aforementioned GC-MS/MS condition. In the previous studies, SIM was the detection mode used for the detection of furan’s derivatives [11,12], however, the detection of a single ion could not completely avoid the interferences from impurities. In this study, not only the MRM mode could be advantageous in more accurate quantification with less interference, but also provided the MRM information for most types of derivatives of furan.

### 2.2. Development of the SPME Arrow Method for Furan Extraction

#### 2.2.1. Evaluation of SPME Arrow Material

Figure 3A presents the adsorption efficacy of the CAR/PDMS, DVB/PDMS, and PDMS arrows, which were evaluated by 20 ppb of furan standard solutions in water as the sample. By comparison, with the exception of furfuryl alcohol, CAR/PDMS exhibited the highest extraction efficiency of furan and its derivatives, followed by DVB/PDMS and PDMS. Interestingly, PDMS showed the highest extraction efficiency of furfuryl alcohol. This outcome also indicated that the adsorption capacity of DVP/PDMS increased following a rise in the molecular weight of analytes. According to a report by Shirey (2012) [14], the pore size in DVP/PDMS was suitable for the adsorption of high-carbon-number compounds, particularly those with a carbon number from 18 to 24. Conversely, CAR/PDMS was the most suitable for analytes with a carbon number from 2 to 16, which was in the same range for the analytes in our study. In a previous study dealing with the determination of furan in commercial food products by using the SPME fibers from PDMS, CAR/PDMS, polyacrylate, PDMS/DVB, carbowax/DVB, and DVB/CAR/PDMS, CAR/PDMS was shown to be the most efficient in furan adsorption [10]. This outcome is consistent with the findings in our study.

#### 2.2.2. Evaluation of Extraction Efficiency by Diluted Solutions

As the extraction efficiency of furfural, furfuryl alcohol, and 2-acetylfuran by water was poor, in our study, we used soybean oil and NaCl solution as a replacement of the diluted solvents. Figure 3B presents the effects of various diluted solvents on the extraction efficiency of furan and its 10 derivatives. The saturated NaCl solution was shown to be superior to the other solvents in extraction efficiency for most of the analytes, possibly due to the enhancement of the ion strength in water after the incorporation of NaCl solution. In addition, the water solubility of analytes may decrease following an increase in the ion strength, resulting in salting-out of the analytes. In a previous study, Altaki et al. [10] also demonstrated that the extraction efficiency of furan was enhanced through a rise in the NaCl level. Comparatively, the extraction efficiency of polar furfural, furfuryl alcohol, and 2-acetylfuran was higher with soybean oil as the diluted medium, probably caused by the low solubility in soybean oil, which, in turn, made them more susceptible to release during the extraction. Nevertheless, compared to water and soybean oil, the saturated NaCl solution was the most optimal for the extraction of furan and its derivatives.

#### 2.2.3. Evaluation of Equilibrium Time

Figure 3C indicates that an equilibrium time of 15 min resulted in the most favorable extraction results, supporting the findings of Condurso et al. [12] and Yu et al. [16].

#### 2.2.4. Evaluation of Ratio of Canned Oily Fish Sample to Saturated NaCl Solution

Our preliminary experiment showed that the extraction efficiency of furan from the canned oily fish samples was inferior to fruit samples, possibly because the oil in the canned fish made furan, slightly liposoluble, making it difficult to separate from the water phase. Thus, the effect of increasing the amount of saturated NaCl solution was explored by using a canned oily fish simulated sample, made from boiled sea bream containing 20% (*w*/*w*) of the soybean oil, and the result is shown in Figure 3D. It was shown that the extraction efficiency of furan was enhanced following an increase in the ratio of saturated NaCl to the sample, and the highest extraction efficiency was attained by mixing 0.5 g of the sample with 9.5 mL of the saturated NaCl solution. However, by taking into account that a small amount of sample (0.5 g) may affect the reproducibility in our study, a sample weight of fish (1 g) and 9 mL of saturated NaCl solution was selected for furan analysis in the fish-based canned sample. In a similar study, a ratio of 1 g of the solid sample to 9 mL distilled water was used, while for the liquid sample, a ratio of 1:1 was employed [17].

#### 2.2.5. The Optimized Method for Analysis of Furan and Its Derivatives

The optimized method for analysis of furan and its derivatives developed by this research was a CAR/PDMS SPME arrow for extraction, a GC system with HP-5MS column for component separation with helium at flow rate 1 mL/min, split mode with split ratio 1:10 and split flow rate 10 mL/min, injector temperature 280 °C, and the following temperature programming conditions: 32 °C in the beginning, maintained for 4 min, raised to 200 °C at 20 °C/min, and maintained for 3 min. A triple-quadrupole tandem mass spectrometer with electron-impact ionization and MRM mode was used for compound detection at an ion source temperature of 230 °C and voltage of 70 eV, while nitrogen was the collision gas at a rate of 1.5 mL/min, and the temperature of mass spectrometer was at 150 °C. The various furan and its 10 derivatives in the samples was detected according to their elution order and the specific m/z as described in Table 1, which shows the operation parameters, such as the precursor ion, as well as the product ion (quantitation ion and confirmation ion), along with their corresponding collision energy used for differentiating each furan and its derivatives.

### 2.3. Method Validation

#### 2.3.1. Recovery

Fresh pineapple, high pressure processed orange juice, and boiled sea bream, containing 20% of soybean oil, were chosen for the blank samples, as the representative of fruit, juice, and canned oily fish samples, respectively. This study investigated the recovery of high-concentration and low-concentration analytes. In the results of Table 2, the recovery was 75.9–114.6% in the canned oily fish matrix, 86.1–113.9% in the fruit matrix, and 84.9–117.2% in the juice matrix. According to the guideline set by the Taiwan Food and Drug Administration (TFDA, 2021) [18], the recovery should be from 50 to 125% when the analyte concentration is ≤0.001 ppm, and from 60% to 125% when the analyte concentration is 0.01–0.1 ppm. Therefore, all the recovery data in our study met the requirement. On the other hand, recovery between 80 to 110% was stipulated for furan and alkyl furans by the Standard Method Performance Requirements (SMPRs^®^) provided from AOAC [19]. Among the data, only one recovery data for 2-pentylfuran at 10 ng/g was lower than 80%, probably due to its lower precision, or because of its higher molecular weight that is not easy to be volatile from the matrix and dilution solution at the extraction temperature of SPME.

In a similar study, Liu and Tsai [20] determined the furan content in juice, canned food, and baby food using SPME-GC-MS and recoveries of 98.9–114.7%, 80.3–105%, and 91.9–108.3% were obtained, respectively. Likewise, Shen et al. [11] used SPME-GC-MS to determine the contents of furan, 2-methylfuran, 2-ehhylfuran, 2,5-dimethylfuran, 2-propylfuran, 2-butylfuran, and 2-pentylfuran in juice and meat products, and recoveries of 88.6–105% and 80–103% were obtained in the meat products and juice, respectively.

#### 2.3.2. Precision

For the canned oily fish sample, the RSD (%) of the intra-day variability was 4-16%, while that of the inter-day variability was 8–19%. Similarly, the RSD (%) of the intraday variability was 1–14%, and that of the inter-day variability was 4-20% for the fruit sample, while the RSD (%) of intra-day variability was 3–11% and that of the inter-day variability was 4–14% for the pineapple sample (Table 2). Based on the TFDA guideline [18], the RSD (%) of the intra-day and inter-day variability must be <35% and <36%, respectively, when the analyte concentration is ≤0.001 ppm; however, when the analyte concentration is 0.01-0.1 ppm, the RSD (%) of the intra-day and inter-day variability must be <30% and <32%, respectively. Therefore, all three matrices used in this study met the requirements set up by the TFDA [18]. In a previous study, Liu and Tsai [20] used SPME-GC-MS to determine the furan contents in juice, canned foods, and baby foods; an RSD (%) of 8.2–20.5%, 3.3–9.2%, and 4.6–10.1% was obtained, respectively, which is comparable to the intra-day and inter-day precision data obtained in our study.

#### 2.3.3. Limit of Detection (LOD) and Limit of Quantitation (LOQ)

The LOD and LOQ for the analytes in the canned oily fish samples were 0.002–0.101 ng/g and 0.007–0.337 ng/g, respectively. However, for the fruit samples, they were 0.001–0.204 ng/g and 0.003–0.675 ng/g, respectively, as well as 0.001–0.048 ng/g and 0.003–0.160 ng/g for the juice samples, respectively (Table 3). By comparison, the juice and canned oily fish samples showed the lowest and highest LOD, respectively. By taking the individual analyte into account, LOQ was the lowest for 2-ethylfuran (0.003–0.01 ng/g) and 2-pentylfuran (0.003–0.007 ng/g), and highest for furfuryl alcohol (0.160–0.675 ng/g) for all samples.

In a study dealing with the determination of the contents of furan, 2-methylfuran, 2-ethylfuran, 2,5-dimethylfuran, 2-propylfuran, 2-butylfuran, and 2-pentylfuran content in juice, ketchup, and meat products by SPME-GC-MS, a low LOD (0.2–0.5 ng/g) was shown for all the analytes in the juice samples [11]. Similarly, Jeong et al. [17] employed a CAR/PDMS SPME fiber to extract furan from commercial food products, fresh fish, and fruit, and reported the LOD for furan to be 0.01 ng/g in the juice and 0.02 ng/g in the canned tuna, probably due to the slightly liposoluble nature of furan and its derivatives, making them easier to separate from the liquid samples. In a similar study conducted by Condurso et al. [12], they used a DVB/CAR/PDMS SPME fiber for the determination of furan and its derivatives in baby foods and reported that the LOD for furan, 2-methylfuran, 2-pentylfuran, 2-butylfuran, 2-acetylfuran, furfural, and furfuryl alcohol ranged from 0.018 to 0.035 ng/g, in which furfuryl alcohol showed the highest LOD (0.029–0.035 ng/g). Perez-Palacios et al. [21] also analyzed the contents of furan and its derivatives in fried fish, and the LOD was found to be the highest for furfuryl alcohol, which can be attributed to its lower sensitivity when detected by MS. By comparison, a much lower LOD was shown in our study, which may partly be due to the use of SPME arrows, resulting in a higher extraction efficiency than the conventional SPME fibers. In addition, a high-resolution power of GC-MS/MS may reduce the interference caused by impurities more efficiently.

According to the US FDA, the concentration of furan in baby foods, infant formulas, coffee, canned vegetable, juice, cereal, and fish were <5.0–108 ng/g, nd~18.8 ng/g, <2.0–84.2 ng/g, nd-122 ng/g, nd-7.6 ng/g, 9.2–47.5 ng/g, and nd-7.1 ng/g, respectively [9]. Moreover, the survey of furan in a UK retail product by the Food Standards Agency (FSA) [22] presented the concentration of furan in breakfast cereals, baby foods, canned olives, and canned prunes as nd-94 ng/g, 5–94 ng/g, 9–16 ng/g, and 5–13 ng/g, respectively, while the 2-methylfuran in those food samples were nd-70 ng/g, nd-11 ng/g, 4–7 ng/g, and nd-1 ng/g, respectively; 3-methylfuran were nd, nd-10 ng/g, 8-9 ng/g, and 2 ng/g, respectively. For 2-ethylfuran, 2,5-dimethylfuran, 2,3-dimethylfuran, furfural, and furfuryl alcohol, the content of the above derivatives in baby food was over 0.3 ng/g [12]. The LOQ presented in our research showed that this method is capable of detecting furan and its derivatives in commercial food samples.

## 3. Materials and Methods

### 3.1. Materials

Since furan could be formed after thermal processing or oxidation, the fresh pineapple, cold-pressed orange juice, and boiled sea bream containing 20% of soybean oil were chosen for the blank samples for method validation, as the representative of the fruit, juice, and canned oily fish samples after the cooking or canning, respectively. All the food raw material were purchased from the local market.

### 3.2. Chemicals and Reagents

The furan standard was procured from Alfa Aesar Co. (Ward hill, MA, USA), while 3-methylfuran was from Acros Co., (Carlsbad, CA, USA), and 2-pentylfuran from Tokyo Chemical Industry Co. (Tokyo, Japan). The other furan derivative standards, including 2-methylfuran, 2,3-dimethylfuran, 2,5-dimethylfuran, 2-ethylfuran, furfural, furfuryl alcohol, 2-butylfuran, and 2-acetylfuran, were from Sigma-Aldrich (St. Louis, MO, USA). The internal standard d4-furan was from Dr. Ehrenstorfer GmbH Co. (Augsburg, Germany). The HPLC-grade methanol was from Merck (Darmstadt, Germany), and the sodium chloride was from Uni Region Bio-Tech (Los Altos, CA, USA). The sodium chloride was purchased from UNI-ONWARD CORP (New Taipei City, Taiwan).

### 3.3. Instrumentation

A 7890B GC coupled with 7000C triple quadrupole tandem mass spectrometer was from Agilent Technologies (Palo Alto, CA, USA). A SPME arrow system with a MPS robotic autosampler was from GERSTEL GmbH & Co. KG (Eberhard-Gerstel-Platz, Germany). Both the HP-WAX (30 m × 0.25 mm, film thickness 0.5 μm) and HP-5MS (30 m × 0.25 mm, film thickness 0.25 μm) columns were also from Agilent Technologies. Three SPME arrows, including the Carboxen/polydimethylsiloxane (CAR/PDMS) SPME arrow (phase thickness 120 μm, outer diameter 1.1 mm), polydimethylsiloxane (PDMS) SPME arrow (phase thickness 100 μm, outer diameter 1.1 mm), and divinylbenzene/polydimethylsiloxane (DVB/PDMS) SPME arrow (phase thickness 120 μm, outer diameter 1.1 mm), as well as the headspace vial with Silicone/PTFE septum, were all from CTC Analytics AG (Zwingen, Switzerland).

### 3.4. Evaluation of Simultaneous Separation of Furan and Its 10 Derivatives by Using GC-MS/MS

A methanol-based standard mixture of furan and 10 derivatives (each at a concentration of 1 ppm) was added to 10 mL of deionized water, for the evaluation of the separation efficiency by both the HP-WAX and HP-5MS columns. A triple-quadrupole tandem mass spectrometer was employed for the detection by electron-impact ionization mode with the ion source temperature at 230 °C and voltage at 70 eV. With multiple reaction monitoring (MRM) mode, the molecular weight of each furan and its derivatives was set as the precursor ion, while nitrogen was the collision gas at a flow rate of 1.5 mL/min for secondary-ion mass spectroscopy. The molecular ion producing the strongest signal was selected to be the quantitation ion, whereas that producing the second strongest signal was chosen to be the confirmation ion. The first and second quadrupoles of the mass spectrometer were maintained at 150 °C.

### 3.5. Development of the SPME Arrow Method of Furan Extraction

#### 3.5.1. Evaluation of the SPME Arrow Material, Diluted Solution and Equilibrium Time

A standard mixture containing furan and its 10 derivatives, each at a concentration of 20 ppb, was added to 10 mL of the distilled water, saturated sodium chloride solution, or soybean oil in the headspace vials and stored at 4 °C. Prior to the GC-MS/MS analysis, all samples were heated at 35 °C for 5, 10, or 15 min, followed by using the SPME arrows made from CAR/PDMS, DVP/PDMS, or PDMS for the adsorption of analytes in the headspace of vials at 35 °C for 15 min. For the equilibrium extraction process, stirring was turned on to assess its influence on the extraction efficiency.

#### 3.5.2. Evaluation of Canned Oily Fish Sample to Saturated NaCl Solution Ratio

The sea bream sample containing 20% (*w*/*w*) soybean oil was homogenized, followed by collecting samples with different weights, 0.5, 1.0, 3.0, and 5.0 g, and mixing with the saturated NaCl solution at volumes of 9.5, 9.0, 7.0, and 5.0 mL, respectively. Then, a standard mixture containing furan and its 10 derivatives, each at a concentration of 20 ppb, was added for storage at 4 °C. Prior to the GC-MS/MS analysis, the samples were heated at 35 °C for 15 min for the subsequent adsorption of analytes by the SPME arrows from CAR/PDMS in the vail headspace at 35 °C for 15 min. The extraction efficiency of furan and its derivatives was affected by the various ratios of fish to the saturated NaCl solution, with obtained base on the GC-MS/MS analysis.

### 3.6. Method Validation

#### 3.6.1. Recovery

Fresh pineapple, high pressure pressed (HPP) orange juice, and boiled sea bream containing 20% (*w*/*w*) soybean oil were used as the blank samples. Specifically, 5 g samples of homogenized pineapple and HPP orange juice were each mixed with 5 mL of the saturated NaCl solution, while 1 g sample of the boiled sea bream containing 20% (*w*/*w*) soybean oil was mixed with 9 mL of the saturated NaCl solution. Then, the samples were mixed with two concentrations (1 and 10 ng/g) of the standards, including furan, 2-methylfuran, 3-methylfuran, 2-ethylfuran, 2,5-dimethylfuran, 2,3-dimethylfuran, 2-butylfuran, 2-acetylfuran, and 2-pentylfuran, separately, as well as 50 and 100 ng/g furfural and furfuryl alcohol, and 2 ng/g of internal standard (d4-furan). Then, the recovery (%) was calculated as follows:


Recovery (%) = [(spiked amount + original amount) − (original amount)]/spiked amount


#### 3.6.2. Precision

The samples including pineapple, HPP orange juice, and boiled sea bream containing 20% (*w*/*w*) soybean oil were spiked in the same way as for the recovery determination, with the exception that the analysis was performed five times on the same day for the determination of the intra-day variability and nine times on three separate days for the determination of the inter-day variability. The relative standard deviation (RSD%) was then calculated for both the inter-day and inter-day variability for the precision evaluation.

#### 3.6.3. Limit of Detection (LOD) and Limit of Quantitation (LOQ)

As mentioned above, GC-MS/MS was performed to obtain both the quantitation ion and confirmation ion signals of furan and its 10 derivatives. Under the autoRMS mode, the baseline signals of the peaks at 0.5 min were treated as noise. Then, the LOD was obtained, based on the signal-to-noise (S/N) ratio ≥3, while LOQ was based on S/N ≥10.

### 3.7. Quantitation of Furan and Its Derivatives

In this study, the standard calibration curves were prepared by spiking the reference standard solutions into blank food matrices. More specifically, 5 g of homogenized pineapple and HPP orange juice were each mixed with 5 mL of the saturated NaCl solution, while 1 g of the boiled sea bream containing 20% (*w*/*w*) soybean oil was mixed with 9 mL of the saturated NaCl solution. Then, the food matrices were spiked with the internal standard and standard of furan and its 10 derivatives, for all the food matrix solutions containing 0.05-6000 ng/g of standards and 20 ng/g of the internal standard. Each standard curve was prepared by plotting the ratio of the analyte peak area to the internal standard peak area, against the ratio of the analyte concentration to the internal standard concentration, and the regression equation and the coefficient (r^2^) of determination was automatically obtained. The following formula was subsequently applied for the calculation:

The concentration of furan and its derivatives (ng/g) = {[(A/RRF)/Ai] × Ci}/weight of sample (g), where A is the peak area for furan and derivatives; RRF is the relative response factor: (A/Ai)/(C/Ci); Ai is the peak area for the internal standard; and Ci is the concentration of the internal standard.

### 3.8. Statistical Analysis

All the analyses were performed in triplicate, and all the data were subjected to ANOVA and Duncan’s multiple range test for the significant difference (*p* < 0.05) in the mean comparison by SAS [23].

## 4. Conclusions

The proposed method involves SPME arrow material, ratio of sample weight with diluted solution, extraction temperature, and equilibrium time; finally, conducting GC-MS/MS analysis were evaluated and developed well in this study. This method was proven capable of facilitating the successful analysis of furan and nine derivatives in various matrices. Moreover, this method has high recovery, a low LOD, a low LOQ, and excellent precision. Its ability to analyze more furan derivatives than other methods within the shortest time and to provide MS/MS ion pairs will enable future studies to analyze furan with greater efficiency and precision.

## Figures and Tables

**Figure 1 molecules-28-01639-f001:**
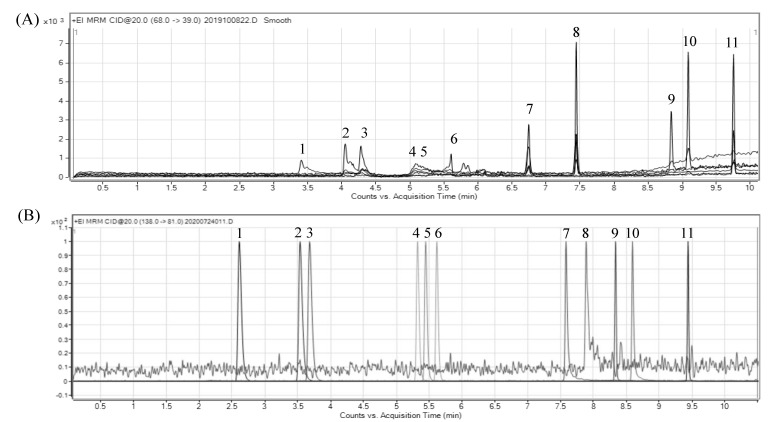
GC−MS/MS chromatogram of furan and its 10 derivative standards by (**A**) HP−WAX and (**B**) HP−5MS columns. Peak identification: (1) furan, (2) 2-methylfuran, (3) 3-methylfuran, (4) 2-ethylfuran, (5) 2,5-dimethylfuran, (6) 2,3-dimethylfuran, (7) furfural, (8) furfuryl alcohol, (9) 2-butylfuran, (10) 2-acetylfuran, (11) 2-pentylfuran.

**Figure 2 molecules-28-01639-f002:**
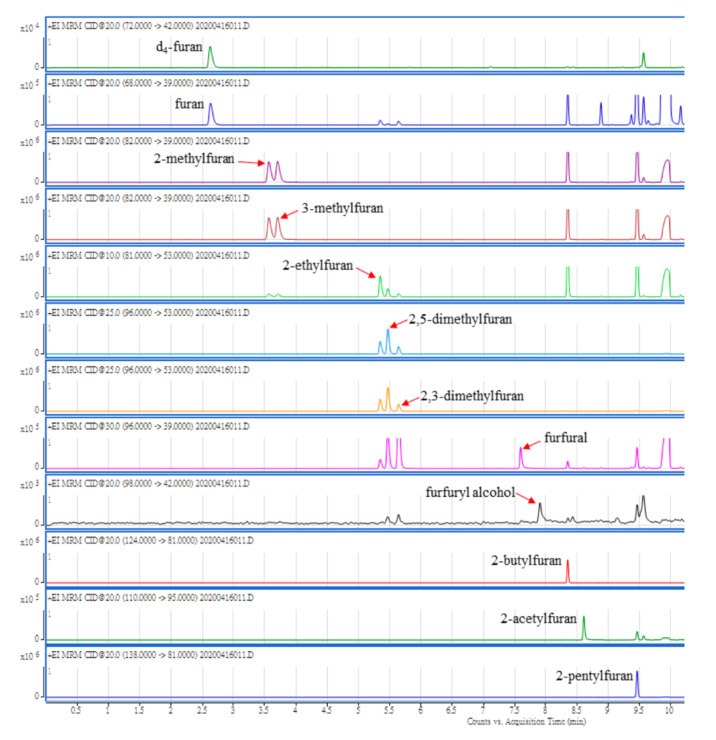
Furan and its 10 derivative standards, detected by GC-MS/MS in MRM mode.

**Figure 3 molecules-28-01639-f003:**
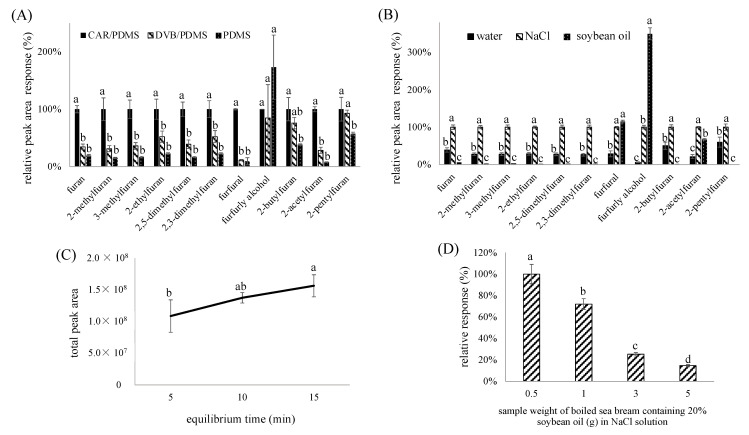
Development of an SPME arrow method for furan and its derivatives extraction. (**A**) Effect of SPME arrow material. (**B**) Effect of diluted solutions. (**C**) Effect of equilibrium time. (**D**) Effect of ratio of oily sea bream weight to saturated NaCl solution. Relative peak area response (%) in y-axis of (**A**,**B**,**D**) were referred to the percentage ratio of the area of each compound extracted by different SPME arrow to the area of each compound extracted by CAR/PDMS. Data with different letters are significant differences at *p* < 0.05. Saturated NaCl solution was used as diluted solution in Figure 3C.

**Table 1 molecules-28-01639-t001:** Operation parameters of furan and its 10 derivative standards, and one internal standard (IS) in multiple reaction monitoring mode by GC-MS/MS.

Peak No.	Compound	Retention Time (min)	Precursor Ion (m/z)	Quantitation		Confirmation	
				Product Ion (m/z)	Collision Energy (V)	Product Ion (m/z)	Collision Energy (V)
IS	d_4_-furan	2.6	72	42	20	44	15
1	furan	2.6	68	39	20	40	20
2	2-methylfuran	3.5	82	39	20	53	20
3	3-methylfuran	3.7	82	39	20	53	20
4	2-ethylfuran	5.3	81	53	10	50	50
5	2,5-dimethylfuran	5.4	96	53	10	81	25
6	2,3-dimethylfuran	5.6	96	53	10	81	25
7	furfural	7.6	96	39	30	68	20
8	furfuryl alcohol	7.9	98	42	20	39	40
9	2-butylfuran	8.3	124	81	20	53	20
10	2-acetylfuran	8.6	110	95	20	39	20
11	2-pentylfuran	9.4	138	81	20	53	35

**Table 2 molecules-28-01639-t002:** Recovery and precision of furan and its derivatives in different food matrix detected by SPME-GC-MS/MS.

Compounds	Spiking Level (ng/g)	Boiled Sea Bream Containing 20% of Soybean Oil			Fresh Pineapple			HPP Orange Juice		
		Recovery ^a^ (%)	Intra-Day Precision (RSD%)	Inter-Day Precision (RSD%)	Recovery (%)	Intra-Day Precision (RSD%)	Inter-Day Precision (RSD%)	Recovery (%)	Intra-Day Precision (RSD%)	Inter-Day Precision (RSD%)
furan	1	106.1 ± 10.5	7%	12%	107.5 ± 2.5	2%	7%	101.7 ± 2.3	4%	9%
	10	109.6 ± 1.2	6%	13%	102.0 ± 2.5	3%	5%	104.1 ± 2.0	6%	7%
2-methylfuran	1	101.7 ± 2.5	4%	10%	106.0 ± 2.4	3%	4%	103.9 ± 0.6	6%	7%
	10	104.3 ± 0.2	12%	13%	110.1 ± 6.6	4%	5%	100.2 ± 2.1	5%	11%
3-methylfuran	1	95.0 ± 9.3	10%	17%	101.7 ± 10.4	6%	12%	103.9 ± 0.6	6%	7%
	10	106.2 ± 2.1	11%	10%	95.6 ± 0.5	6%	7%	100.2 ± 2.1	5%	11%
2-ethylfuran	1	101.4 ± 5.2	9%	16%	111.4 ± 1.8	1%	10%	87.2 ± 3.3	6%	6%
	10	103.5 ± 7.2	16%	13%	108.0 ± 1.8	2%	8%	96.2 ± 1.2	3%	5%
2,5-dimethylfuran	1	88.3 ± 1.3	8%	15%	113.9 ± 2.5	4%	11%	93.1 ± 1.8	9%	11%
	10	102.5 ± 2.7	9%	11%	105.4 ± 1.2	5%	10%	102.4 ± 0.6	3%	4%
2,3-dimethylfuran	1	88.3 ± 1.3	10%	10%	113.9 ± 2.5	2%	5%	93.1 ± 1.8	7%	14%
	10	106.4 ± 4.2	9%	10%	103.1 ± 0.8	2%	10%	90.7 ± 6.0	3%	10%
furfural	50	100.0 ± 2.6	9%	10%	103.2 ± 11.8	12%	20%	96.8 ± 3.8	5%	9%
	100	98.8 ± 14.7	7%	16%	100.9 ± 11.4	14%	15%	89.2 ± 1.8	9%	7%
furfuryl alcohol	50	94.9 ± 2.5	10%	13%	104.1 ± 3.1	6%	7%	108.2 ± 6.0	10%	13%
	100	82.1 ± 5.5	12%	15%	109.6 ± 6.3	6%	8%	84.9 ± 2.0	4%	5%
2-butylfuran	1	87.8 ± 9.7	11%	13%	92.2 ± 6.4	6%	9%	115.7 ± 4.1	4%	10%
	10	82.1 ± 5.5	11%	13%	109.6 ± 6.3	7%	12%	84.9 ± 2.0	3%	10%
2-acetylfuran	1	114.6 ± 4.1	10%	17%	86.1 ± 4.9	10%	16%	86.9 ± 4.8	7%	9%
	10	96.0 ± 0.1	5%	8%	98.5 ± 8.0	8%	10%	117.2 ± 0.4	8%	8%
2-pentylfuran	1	87.34 ± 2.1	11%	19%	98.1 ± 17.4	9%	7%	93.1 ± 1.8	11%	13%
	10	75.92 ± 0.7	10%	14%	103.8 ± 7.5	8%	9%	112.9 ± 5.3	7%	12%

^a^ recovery (%) = (amount found–original amount)/amount spike * 100%. Average of double analysis ± standard deviation.

**Table 3 molecules-28-01639-t003:** Limit of detection (LOD), limit of quantitation (LOQ) of furan, and its derivatives in different food matrix detected by SPME-GC-MS/MS.

Compounds	Boiled Sea Bream Containing 20% Soybean Oil		Fresh Pineapple		HPP Orange Juice	
	LOD ^a^ (ng/g)	LOQ ^b^ (ng/g)	LOD (ng/g)	LOQ (ng/g)	LOD (ng/g)	LOQ (ng/g)
furan	0.024	0.080	0.004	0.014	0.006	0.020
2-methylfuran	0.022	0.073	0.001	0.004	0.001	0.003
3-methylfuran	0.022	0.073	0.019	0.063	0.001	0.003
2-ethylfuran	0.003	0.010	0.001	0.003	0.001	0.003
2,5-dimethylfuran	0.049	0.163	0.006	0.022	0.003	0.010
2,3-dimethylfuran	0.094	0.313	0.002	0.006	0.007	0.023
furfural	0.012	0.040	0.005	0.019	0.004	0.013
furfuryl alcohol	0.101	0.337	0.204	0.675	0.048	0.160
2-butylfuran	0.011	0.037	0.001	0.003	0.001	0.003
2-acetylfuran	0.038	0.127	0.013	0.043	0.015	0.050
2-pentylfuran	0.002	0.007	0.001	0.003	0.001	0.003

^a^ LOD based on S/N = 3 of food matrix. ^b^ LOQ based on S/N = 10 of food matrix.

## Data Availability

The data presented in this study are available in this article.

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
