# Peer review of "Analysis of Furan and Its Derivatives in Food Matrices Using Solid Phase Extraction Coupled with Gas Chromatography-Tandem Mass Spectrometry"

_molecules, 2023, doi:10.3390/molecules28041639_

Round 1

Reviewer 1 Report

In this manuscript, the authors have developed SPE-GC-MS/MS method for the analysis of furan and its derivatives in various food samples.

The work presented here is interesting but there are some shortcomings in it which should be addressed before acceptance and publication in Molecules.

The comments are given in the word file attached here 

Reviewer 2 Report

The paper is devoted to the quantitative and qualitative determination of furane and its derivatives in food. The area of the investigation is very important due to necessity of controls of these compounds in very low levels. The authors discuss in the paper the method of simultaneous determination of furan and its ten derivatives in different food samples. The authors demonstrated that furan and its derivatives might be separated rather fast using 5% phenyl column and GC-MS/MS in MRM mode for detection.

There are following questions and remarks concerning the paper:

1) Separation of furan and its derivatives using 5% phenyl GC columns of different manufacturers is already described in the literature, see, for example, DOI: 10.1007/s00217-020-03556-2. There is nothing new in this part.

2) The lower part of the chromatogram is cutted in Fig. 1(B)? Why are the peak heights are so different in the chromatogram for peak 8 and all the others? The information on the concentration of the standard solutions should be added.

3) Please, correct Fig. 3, it is unreadable.

4) Why the boiled fish was used as the food sample, if it is known that furan and its derivatives are low-boiling compounds, and their quantities may be significantly reduces during the procedure?

5) The conclusions should be reformulated. The conclusions in the present form are more likely the abstract.

6) The half of the references in the literature are ten years and elder.

Reviewer 3 Report

This paper developed a simultaneous analysis method of furan and its 10 derivatives in different food commodities. The method validation data demonstrated that a reliable method was established for analysis of furan and its 10 derivatives in commercial foods. However, there are some weaknesses that need to be addressed.

Line 90: delete the word “8”.

Line 110: Suggest to change “adequately separation” to “adequately separated”.

Line 115-116: what are the meanings of the number 2 and 1?

Line 140-140: The table title should be above the table.

Figure 3:

(1) We cannot clearly find the data from the images. The figure 3 should be changed with high definition and correct formatting.

(2) The significant differences expressed in letters (A-D) in figure 3 were better to use lowercase letters (a-d) instead.

Line 249: From Table 2, the RSD (%) of the intra-day variability was 4-16% for the canned oily fish sample.
